# Testing Strategies of the In Vitro Micronucleus Assay for the Genotoxicity Assessment of Nanomaterials in BEAS-2B Cells

**DOI:** 10.3390/nano11081929

**Published:** 2021-07-27

**Authors:** Tereza Cervena, Andrea Rossnerova, Tana Zavodna, Jitka Sikorova, Kristyna Vrbova, Alena Milcova, Jan Topinka, Pavel Rossner

**Affiliations:** 1Department of Nanotoxicology and Molecular Epidemiology, Institute of Experimental Medicine CAS, Videnska 1083, 142 20 Prague, Czech Republic; tereza.cervena@iem.cas.cz (T.C.); kristyna.vrbova@iem.cas.cz (K.V.); 2Department of Physiology, Faculty of Science, Charles University, 128 44 Prague, Czech Republic; 3Department of Genetic Toxicology and Epigenetics, Institute of Experimental Medicine CAS, Videnska 1083, 142 20 Prague, Czech Republic; andrea.rossnerova@iem.cas.cz (A.R.); tana.zavodna@iem.cas.cz (T.Z.); jitka.sikorova@iem.cas.cz (J.S.); alena.milcova@iem.cas.cz (A.M.); jan.topinka@iem.cas.cz (J.T.)

**Keywords:** nanomaterials, micronucleus assay, genotoxicity, cell line, DLS

## Abstract

The evaluation of the frequency of micronuclei (MN) is a broadly utilised approach in in vitro toxicity testing. Nevertheless, the specific properties of nanomaterials (NMs) give rise to concerns regarding the optimal methodological variants of the MN assay. In bronchial epithelial cells (BEAS-2B), we tested the genotoxicity of five types of NMs (TiO_2_: NM101, NM103; SiO_2_: NM200; Ag: NM300K, NM302) using four variants of MN protocols, differing in the time of exposure and the application of cytochalasin-B combined with the simultaneous and delayed co-treatment with NMs. Using transmission electron microscopy, we evaluated the impact of cytochalasin-B on the transport of NMs into the cells. To assess the behaviour of NMs in a culture media for individual testing conditions, we used dynamic light scattering measurement. The presence of NMs in the cells, their intracellular aggregation and dispersion properties were comparable when tests with or without cytochalasin-B were performed. The genotoxic potential of various TiO_2_ and Ag particles differed (NM101 < NM103 and NM302 < NM300K, respectively). The application of cytochalasin-B tended to increase the percentage of aberrant cells. In conclusion, the comparison of the testing strategies revealed that the level of DNA damage induced by NMs is affected by the selected methodological approach. This fact should be considered in the interpretation of the results of genotoxicity tests.

## 1. Introduction

Nanomaterials can be defined as “a natural, incidental or manufactured material containing particles, in an unbound state or as an aggregate or as an agglomerate and where, for 50% or more of the particles in the number size distribution, one or more external dimensions is in the size range 1–100 nm” [1]. A tremendous increase in their application has been witnessed during the last decade. In Europe alone, over 5000 registered products can be purchased with various nanomaterials (NMs) used, as evident from Nanodatabase (www.nanodb.dk; accessed on 26 July 2021), an online inventory of products, which can either be bought in Europe or ordered online and shipped to a European location. Nanowerk Nanomaterial Database (www.nanowerk.com; accessed on 26 July 2021), another database, lists more than 4500 types of nanomaterials. Due to possible human exposure via a large number of consumer products containing NMs, the interest of toxicologists in these materials is steadily increasing [2].

Current evidence shows that exposure to NMs has numerous toxic effects in both prokaryotes and eukaryotes. In general, in vivo and in vitro studies in model systems suggest that NMs induce an inflammatory response, DNA damage, oxidative stress, lipid peroxidation, apoptosis, altered gene expression, cytotoxicity and reproductive toxicity [3,4,5,6,7,8]. Despite this fact, there is still a limited number of human biomonitoring studies focusing on the effects of NMs. Studies investigating the biological impacts of occupational exposure to titanium dioxide, iron oxide or multi-walled carbon nanotubes, observed elevated oxidative damage to DNA, lipids and proteins in exhaled breath condensate [9,10] or elevated inflammatory cytokines in sputum samples [11]. Most recently, differences in DNA methylation pattern, as well as DNA damage after acute and chronic occupational exposure to NMs, were reported [12,13]. Interestingly, the application of cytogenetic methods that allow the detection of DNA breaks, losses and/or rearrangements is rare in the human biomonitoring of NMs exposure. The investigation of the total micronuclei (MN) by a cytokinesis-block micronucleus assay (CBMN assay) in binucleated cells (BNC) is an established, traditional method [14], successfully used for the evaluation of the effect of exposure to many chemicals [15,16]. However, to the best of our knowledge, the results from only one human study focused on MN formation following the exposure to NMs was reported [17,18]. In addition, one study evaluated the ex vivo effect of TiO_2_ nanoparticles (NPs) exposure in the lymphocytes of three human subjects [19].

In contrast, in vitro studies utilising a broad spectrum of cell lines, and evaluating the frequency of MN after NMs exposure, are relatively common. However, this field of research raises several issues, particularly due to the number of various types and unique physico-chemical properties of NM (e.g., size, shape or aggregation) that are the key factors in the experimental design. These properties are responsible for the unexpected interactions with experimental components, supporting the fact that NMs cannot be treated in the same manner as chemical compounds [20]. Choosing the optimal methodological variant of the micronucleus assay, respecting e.g., the possible transport mechanism of NMs into the cells, is a major task of the genotoxicity testing that affects the interpretation of the results [21,22]. Few studies have discussed the potential methodological issues, such as the timing of an experiment, the presence and concentration of serum in a culture media, as well as the application and concentration of cytochalasin-B (CB). Despite the above-mentioned facts, the majority of studies do not take the design of MN experiments into account. Particularly, there are doubts about the application of CB and the strategy of NM toxicity testing. The main reason for this debate is the fact that CB is an actin filaments inhibitor. If added to the cell culture, cytoplasmic cell division is stopped, and the process of actin-dependent endocytosis is affected. This can lead to the underestimation of genotoxic potential, as actin-dependent endocytosis is one of the routes of entry for NMs into the cells [16].

In this study, we considered data previously published by us and other authors and concentrated on the knowledge gaps in optimising the micronuclei assay for the toxicity tests of a broad range of NMs. We selected five types of well-characterised NMs (TiO_2_: NM101 and NM103, SiO_2_: NM200, Ag: NM300K and NM302) that represent commonly used NM groups. The NMs differed by many characteristics, including their application, size, shape and aggregate dimension. We used four variants of MN protocols characterised by the time of exposure, and the application of CB combined with simultaneous or delayed co-treatment with NMs. Using transmission electron microscopy (TEM), we also evaluated the possible impact of CB on the reduced or blocked the transport of NMs into the cells. Finally, to assess the effect of CB presence on NMs behaviour in a culture media, we performed dynamic light scattering (DLS) measurement.

## 2. Materials and Methods

### 2.1. Preparation of NMs

Five types of NMs differing by their characteristics were selected for the experiments: (i) TiO_2_ [titanium dioxide—anatase form, primary particle size 5 nm, NM101 (Joint Research Centre)]; (ii) TiO_2_ [titanium dioxide—rutile form, primary particle size 20–100 nm, NM103 (Joint Research Centre)]; (iii) SiO_2_ [synthetic amorphous silicon dioxide produced by precipitation, particle size 10–20 nm, NM200 (Joint Research Centre)]; (iv) Ag [silver with prevalently round shape, particle size < 20 nm, dispersion 100 mg/mL, NM300K (Joint Research Centre)]; and (v) Ag [silver with rods shape, particle size 100–200 nm width, 5–10 µm length, 91.26 mg/mL, NM302 (Fraunhofer) [23,24,25,26]]. The basic characteristics of NMs are summarised in Table 1.

The handling of all tested NMs respected the above-cited manufactured protocols. The preparation of NMs was performed according to the instructions of the generic NANOGENOTOX dispersion protocol (www.nanogenotox.eu; accessed on 26 July 2021) [27]. Briefly, 15.36 mg of NMs were weighed in glass vials using a microbalance (RC210D, Sartorius, Germany). The NMs were then pre-wetted with 30 µL of 95% ethanol and diluted in 5.97 mL deionised water (diH_2_O) containing 0.05% bovine serum albumin (BSA, Sigma-Aldrich, St. Louis, MO, USA). If the NMs were available in a dispersion form, the amount of dispersion used was calculated to correspond to 15.36 mg of dry NMs. In total, 6 mL suspension of NMs in 0.05% BSA had a concentration of 2.56 mg/mL (batch). Batches were sonicated (400 W, 10% amplitude, 3136 MJ/m^3^) by an ultrasonic homogeniser (S-450d, Branson, MO, USA), equipped with a standard 13-mm disruptor horn for 16 min in an ice bath, to prepare batch suspensions at a concentration of 2.56 mg/mL [27]. The homogeniser was calibrated according to NANOGENOTOX dispersion protocol. The suspension was vortexed for 2 min before manipulation and gradually diluted by BEGM™ culture medium to reach final concentrations of 25, 10, and 1 µg/mL for NM101, NM103, and NM302 or 5, 2.5, and 1 µg/mL for NM300K and NM200. NMs were used for DLS analysis and cell exposure within 1 h.

### 2.2. Characterisation of NMs

Properties of the NM dispersions (average hydrodynamic size [nm], polydispersity index (PDI); zeta-potential [mV]) were measured after preparation in diH_2_O containing 0.05% BSA (batch), or in BEGM™ culture medium, at 0 h, 28h and 48 h (NMs diluted to tested concentrations). As CB was also used in the MN assay, we performed NM dispersion characterisation with and without CB (1 µg/mL).

The hydrodynamic size in suspensions was measured using DLS analysis (Zetasizer Nano ZS, Malvern, UK). Hydrodynamic size (Z-Avg) and polydispersity index (PDI) were determined according to the ISO method (ISO13321:1996, ISO22412:2008). Batch suspensions in diH_2_O containing 0.05% BSA were measured within 20 min after sonication. Medium suspensions were analysed at 0 h, 28 h and 48 h after preparation. All suspensions were vortexed before analysis. Between the measurement time points, medium suspensions were kept in the incubator (37 °C, 5% CO_2_) to simulate the same conditions as during toxicity testing. Diluted samples were measured in polystyrene cell cuvettes (DTS0012), stabilisation time was set to 120 s, temperature for batch to 25 °C, and 37 °C for medium suspensions. Repeated runs (12–24) were performed for each time and concentration. The same settings were used for measuring zeta potentials (DTS1070 cuvettes).

### 2.3. Cell Cultivation and Treatment Conditions

Human bronchial epithelial cells BEAS-2B (CRL-9609^TM^, ATCC^®^, Manassas, VA, USA) were used to evaluate the genotoxic effect of the selected NMs. The BEAS-2B cells represent an adherent cell line derived in 1988 from the lung autopsy of a healthy man [28]. The cells are non-tumour, immortalised by the hybrid Ad12-SV40 virus, with standard morphology and metabolism. The cells are pseudodiploid and stable under conditions defined by ATCC^®^.

The cultivation protocol recommended by ATCC^®^ cells was used in this study, in order to ensure the reproducibility of the experiments. Briefly, cell cultivation surfaces were coated with a mixture of 0.01 mg/mL fibronectin (Sigma-Aldrich, St. Louis, MO, USA), 0.03 mg/mL bovine collagen type I (Sigma-Aldrich, St. Louis, MO, USA) and 0.01 mg/mL BSA (Sigma-Aldrich, St. Louis, MO, USA) dissolved in a bronchial epithelial basal medium (BEBM™, Lonza, Basel, Switzerland) and kept in a 37 °C incubator overnight. Before seeding the cells, all the coating media were removed. Serum-free cultivation conditions (BEGM™ kit CC3170) (Lonza, Basel, Switzerland) were used. For all the experiments, the confluence of the cells did not exceed 70% to avoid terminal squamous differentiation.

### 2.4. Micronucleus Analysis

The genotoxicity of the selected NMs was determined using the micronucleus assay in both mononucleated and binucleated cells [29,30]. We selected three non-toxic concentrations for each NM, based on cytotoxicity assays WST-1 and MTT conducted as described in [31]. TiO_2_ NM-101, NM-103 and Ag NM-302 were tested at 1 µg/mL, 10 µg/mL and 25 µg/mL (i.e., 0.25 µg/cm^2^, 2.5 µg/cm^2^ and 6.25 µg/cm^2^). Ag NM-300K and SiO_2_ NM-200 were tested at 1 µg/mL, 2.5 µg/mL and 5 µg/mL (i.e., 0.25 µg/cm^2^, 0.625 µg/cm^2^ and 1.25 µg/cm^2^). As a negative control, we used 0.1% DMSO (binucleated samples) or diH_2_O (mononucleated samples). BEAS-2B cells incubated with benzo[a]pyrene (25–200 µM) served as a positive control (data not shown). The cells were treated for 28 h or 48 h in one exposure experiment in triplicate cultures using the same passage of the cells.

Four methodological variants, two without CB and two with CB, were selected for testing in this study. The 8-well Lab-Tek™ Chamber Slide System was used for cell cultivation and treatment to reduce the consumption of media and the cost of the experiments. For the evaluation of mononucleated cells, NMs were applied for 28 h or 48 h (no treatment with CB strategy—var. 1 and var. 2, respectively). In experiments focused on the analysis of the binucleated cells, the co-treatment version of cytokinesis-block micronucleus assay with simultaneous incubation with nanomaterials and CB (var. 3) and delayed co-treatment with the addition of CB after nanomaterials treatment (var. 4), for 28 h or 48 h, respectively, was performed according to the previously described protocol [21,32]. The concentration of CB (Sigma-Aldrich, St. Louis, MO, USA) was 1 µg/mL. An overview of all the considered methodological variants (4 used and 2 not applied in the study) is shown in Figure 1A–F (see section “Results” for more details and reasons for this selection). At the end of cultivation, the cells were treated with a hypotonic solution of KCl (0.075M, Sigma-Aldrich, St. Louis, MO, USA) and fixed with a mixture (3:1) of methanol (Merck Millipore, Billerica, MA, USA) and acetic acid (Penta, Prague, Czech Republic).

After fixation, the slides were dried and stained with 5% Giemsa (Merck Millipore, Billerica, MA, USA). Visual scoring using the Olympus BX41 microscope was performed to analyse the mononucleated cells (MONO) or binucleated cells (BNC), at a final magnification 1000×. A total of 3× 500 BNC and 3× 1000 MONO for each tested compound were evaluated. For BNC analysis, the cytokinesis-block proliferation index (CBPI) was calculated to control for cell division according to the following formula [33]:[(number of mononucleated cells + 2 × number of binucleated cells + 3 × number of tri- and tetranucleated cells/500 (a total cell number)] 

The aberrant cells were recorded using a Canon EOS600D camera. Examples of BNC cells with tested NMs are shown in Figure 2A–C. The results were expressed as a percentage of aberrant mononucleated or binucleated cells, with micronuclei (% ABB) of a total number of scored cells.

### 2.5. Transmission Electron Microscopy

Transmission electron microscopy (TEM) was used to check the presence of NMs and their agglomerates inside the cells and/or nuclei. BEAS-2B cells were incubated with the NMs of selected concentrations (NM-101, NM-103, NM-302: 25 µg/mL; NM-300K and NM-200: 5 µg/mL) for 28 h, washed with HEPES buffer (0.1 M, pH 7.2) at 37 °C, fixed with 2.5% glutaraldehyde in HEPES for 1 h, washed, and postfixed with 1% OsO_4_ solution in HEPES for 45 min. The samples were dehydrated in an ethanol gradient, followed by propylene oxide, and embedded in Quetol 651 resin. After polymerisation for 72 h at 60 °C, blocks were cut into 80 nm ultrathin sections and collected on 200 mesh size copper grids. The sections were examined using the FEI Morgagni 268 transmission electron microscope operated at 80 kV. The images were captured using the Mega View III CCD camera (Olympus Soft Imaging Solutions). For each NM, three samples for TEM were prepared and analysed.

### 2.6. Statistical Analysis

The two-proportion z-test, to compare control groups versus treated groups, was used to analyse MN data. The differences between the groups were considered significant for *p* < 0.05.

## 3. Results

### 3.1. Characterisation of NMs

The properties of NM dispersions (average hydrodynamic size [nm], polydispersity index (PDI), zeta-potential [mV]) were measured after preparation in diH_2_O containing 0.05% BSA (batch) (Table 2), and in BEGM™ culture medium at 0 h, 28 h and 48 h after preparation. The NMs characterisation in BEGM™ culture medium was done with and without CB (1 µg/mL) (Table 3 and Table 4, respectively).

Among the tested NMs, both titanium dioxide NMs were the most stable (regardless of the presence of CB), and their average hydrodynamic size did not substantially change over time, although NM101 at the lowest concentration (1 ug/mL) applied without CB increased in size after 28 h and 48 h. Additionally, NM200 (regardless of the presence of CB) and NM300K (without CB) had a similar average hydrodynamic size at time zero when compared to the batch (Table 2, Table 3 and Table 4, Figure 3, Figure 4, Figure 5, Figure 6 and Figure 7). After 28 h and 48 h, we observed a size increase for NM200, NM300K and NM302. This observation, together with higher PDI, indicates the formation of aggregates of NMs in the BEGM™ over time. The size change was reduced by the application of CB (Appendix A). According to the manufacturer [34], the DLS measurement is only suitable for the suspension with PDI < 0.7. When the PDI is greater, the suspension is too polydisperse, and the results are not guaranteed. In addition, the method is not recommended for nanorods or nanowires due to their shape (NM302). Indeed, for NM302 (and NM200) the value exceeded the limit in some cases (indicated in Table 3 and Table 4), and thus the average hydrodynamic results are not reliable for these samples. Those NMs aggregated rapidly in the medium and the cells would be most likely exposed to large aggregates of these NMs, rather than to the small clusters or individual particles (Figure 6 and Figure 7). The more stable NMs (NM100 and NM103) showed no or slight size growth with increasing cultivation time (Figure 3 and Figure 4). The higher values of PDI were mainly present after 48 h of cultivation. Cells exposed to NM100 and NM103 would be more likely to be in contact with smaller clusters and aggregates of NMs than larger ones. The Zeta potential for the tested NMs varied prevalently between −10 and −17 mV, and mostly did not change with cultivation time (maximum of −9.94 ± 1.25 mV (NM300K, 5 µg/mL 48 h with CB) and minimum of −21.11 ± 1.64 (NM300K, 5 µg/mL 48 h without CB)). The Zeta potential of absolute values between 10 and 20 mV suggests a relatively stable suspension [35,36].

### 3.2. Transmission Electron Microscopy

The presence of the tested NMs in BEAS-2B cells was confirmed by TEM images (Figure 8A–K). Although exact quantification was not performed, the increased accumulation of titanium dioxide NMs (NM101 and NM103) was visible. This fact complicated the TEM sample preparation (visible as sample tears). A lower number of SiO_2_ particles was found in the cell cytoplasm, but their presence was confirmed in samples prepared both with and without CB. Silver NMs (NM300K, and particularly NM302) were the least frequently found NMs in the cells, although their presence was confirmed in both the CB positive and negative samples. NM302 (silver rods, particle size 100–200 nm width, 5–10 µm length) was detected as small, elongated fragments. Although we were not able to determine the exact number of NMs in BEAS-2B cells, we confirmed their presence in all the treated samples, regardless of the application of CB.

### 3.3. Micronucleus Analysis

The studied MNs were used in four variants of the MN protocol, differing in the time of exposure (28 h or 48 h) and application of CB. A total of six different approaches were originally considered (Figure 1A–F) but the protocols that involved washing steps (Figure 1C,F) were excluded, as uniform effectiveness of washing after the NMs exposure could not be guaranteed. In methodological variants with CB, we used simultaneous or delayed co-treatment with NMs. The results, summarised in Figure 9A–E, are expressed as a percentage of aberrant cells (% ABB) among mononucleated or binucleated cells.

Our data revealed a difference between the effects of two types of tested TiO_2_ NMs (NM101 and NM103). The frequencies of MN in the cells exposed to NM101 (Figure 9A) did not significantly increase above the negative control level in both the samples processed without (Figure 1A,B) and with CB (Figure 1D,E). In contrast, the samples exposed to NM103 using the same exposure scheme showed elevated frequencies of MN after 28 h of simultaneous NMs and CB treatment. This result was obtained for the lowest and highest tested concentrations (Figure 9B). Similar results were observed after exposure to SiO_2_ (NM200), although a 48 h delayed co-treatment with CB and NMs was needed to observe the significant increase of MN frequencies. For Ag (NM-300K), MN were formed after the application of both protocols. However, the approach without CB required exposure to higher doses and a 48 h treatment with NMs (Figure 9C,D). Interestingly, MN results after exposure to Ag NM302 show only minor, non-consistent, differences between the use of MN assay protocol with or without CB.

## 4. Discussion

Numerous papers addressing the genotoxicity of NMs investigated by one of the variants of the in vitro micronucleus assay have been published [37,38]. The majority of them have focused on the various size fractions of titanium dioxide (a possible carcinogen to humans according to the International Agency for Research on Cancer [39], group 2B), one of the most common NMs used in industry. Its worldwide production is about 4 million tons, of which approximately 3000 tons is produced in NM form [40]. The wide range of results obtained by micronucleus assay, after the treatment of the cells with various mixtures of both crystalline forms (anatase and rutile) of TiO_2_ differing by their size, show a span from negative to positive effects. No impact of exposure was observed for 100% anatase TiO_2_ > 200 nm in BEAS-2B cells, in contrast to smaller 10 and 200 nm NMs, which induced a significant increase in the frequency of MN [41]. No consistent results were observed for the different mixtures of both crystalline forms in peripheral blood lymphocytes [42] and human hepatocytes [43]. Genotoxic/transforming effects were observed in another study, only after the application of a high, non-realistic dose of tested NMs (1000 mg/mL) [44]. The effect of size was also observed for SiO_2_ NPs, where smaller crystalline forms (7.21 nm) caused a significant increase of MN frequency in contrast to larger amorphous forms (up to 104 nm) [45,46]. Ag NPs in the size range 6–20 nm induced MN in two cell lines [47], and affected the cytotoxicity and genotoxicity in a size- and coating-dependent manner [48]. The size effect of NMs on the resulting cytotoxicity and/or genotoxicity level was also presented by other authors [49,50,51].

Various methodological confounding factors may affect the results of the genotoxicity assessment of NMs when using the MN assay [52]. Among them, the experimental design of the MN test that either includes or omits CB seems to be fundamental. Several MN assay methodologies were previously discussed for dextran-coated USPION exposure to MCL-5 cells (in 1% serum), where a post-treatment with CB induced a significant and dose-dependent increase of MN frequency [21]. A similar approach, that also included delayed co-treatment with CB, was suggested in later studies [20]. Another concern is the presence of serum proteins in the culture media, which has been shown to modulate the cytotoxic and genotoxic responses to NMs. This factor should be carefully considered as it possibly changes the sensitivity of various assays [53,54,55]. The selection of a cell line is also crucial, as a mutation in the *p53* gene may increase sensitivity to MN induction by NMs [52]. An understanding of the confounding factors can be valuable in the interpretation of genotoxicity results. Moreover, the type of MN analysis (visual or flow cytometry) must be selected with respect to the type of analysed NMs.

The application of CB is a basic methodological variant of the MN assay. A risk of underestimation of NMs genotoxicity, assessed as MN frequency, linked with potential inhibition of endocytosis by CB was previously discussed [16]. Verification of this possible limitation was the main aim of our study. A total of six MN protocols shown in Figure 1 were originally considered in our experiments. However, we decided to exclude the variants with washing followed by post-treatment cultivation without and with CB (Figure 1C,F), as the effectivity of these steps may vary between individual experiments. Our results confirmed that MN are formed even if genotoxic effects are analysed in the presence of CB. Consistently comparable results for all NM concentrations and both variants of the MN test were obtained for TiO_2_ NM101 treatment. An even higher frequency of MN for the variant with CB was detected for some doses of TiO_2_ NM103, Ag NM 300K and SiO_2_ NM200. Moreover, the extension of the exposure time to 48 h in the CB variant caused an additional MN frequency increase induced by the SiO_2_ NM200 treatment. Even though some differences in the data were observed for individual protocols, there is no evidence that the results were underestimated. Although CB plays a crucial role in the blocking of the process of cytokinesis, the formation of MN does not seem to be affected. Four scenarios of endocytosis were described (phagocytosis, pinocytosis, clathrin-mediated endocytosis and caveolae formation) for the transport of particles of various size (from nm to µm) into the cells. The existence of these alternative approaches can explain why NPs enter the exposed cells also in the presence of CB. This is supported by a study investigating preferences of mechanisms of endocytosis of multiwalled carbon nanotubes in bronchial epithelial and mesothelial cells by clathrin-mediated or caveolae-mediated pathways [56].

The DLS data showed differences in the stability of the tested NMs under various conditions, with NM103 (titanium dioxide–rutile form, primary particle size 20–100 nm) being the most stable. As expected, the NMs behaved differently in dispersion over time, and their nominal size determined by the manufacturer has changed. Overall, after 28 h and 48 h, we observed an increase in size for NM200, NM300K and NM302; TiO_2_ NMs were the most stable. Interestingly, a higher or comparable increase in the average hydrodynamic size [nm] was observed for samples without CB compared to the CB samples (Appendix A). These observations, together with higher PDI, indicate the formation of aggregates in the BEGM™ over time. Supporting our results, the time-dependent stability of TiO_2_ was also observed in previously published work on TiO_2_ in THP-1 cells [31]. The effect of culture medium on NMs stability and their in vitro effects was previously discussed e.g., by Precupas et al. [57] and Prasad et al. [58]. Further, our unpublished data showed that LHC-9, an alternative culture medium recommended for BEAS-2B cells that is characteristic of high protein agglomeration over time, negatively impacted stability of NMs at a concentration below 25 µg/mL. These results suggest that different reactions to the NMs in various in vitro models can be influenced by culture medium.

According to TEM images, BEAS-2B cells engulfed all types of tested NMs. The highest accumulation of NMs inside the cells was observed in samples treated with TiO_2_ (NM101 and NM103). Interestingly, some cells were loaded with dozens of agglomerates resulting in problems with TEM sample preparation. NMs were predominantly found in membrane-enclosed cellular compartments indicating active cell transport into the cells such as endocytosis. Interestingly, smaller agglomerates and even clusters of few nanoparticles were also found in some cases (Figure 8I).

The extensive testing of NMs that was carried out in this study was inspired by previously published reports, and the unique physico-chemical properties of NMs responsible for potentially unexpected interactions with experimental components [20]. Additionally, a recent review focused on an alternative testing strategy of NMs discussed questions concerning endocytosis blocking after the application of CB [59]. Our results indicate that the presence of CB affects the MN assay; stronger effects were observed for TiO_2_ (NM103), SiO_2_ (NM200) and Ag (NM300K). This observation might be related to specific physico-chemical characteristics of these NM, resulting in, for example, better visibility of MN induced by these NM. The testing strategies omitting CB represent a situation typical for the real exposure of humans. Co-treatment with CB can only simulate the possible reaction of the human organism during cancer therapy. Concerning other findings, two types of TiO_2_ NMs (NM101 and NM103) and Ag NMs (NM300K and NM302), which differed in size and crystalline form or shape, respectively, are typical examples of the differences in the genotoxic effect caused by the properties and not by the material that they are made from. For the results obtained for NM101 and NM302, we may hypothesise that the simultaneous treatment of BEAS-2B cells with NMs and CB does not lead to an underestimation of their genotoxic potential. This may be due to the alternative pathway/s of transport through the cell membrane and, e.g., the solubility of Ag NPs [22].

## 5. Conclusions

In conclusion, particles and their aggregates of different size were present in BEAS-2B cells after the treatment with all tested NMs, independently of the absence or presence of CB during cultivation. The DLS results showed a partial time-dependent increase of average hydrodynamic size [nm] for all the tested NMs of at least one tested concentration, except for NM103. The addition of CB reduced this increase for NM101, NM300K and NM200. Despite the DNA damage level measured by micronucleus assays without blocking of cytokinesis being relatively insignificant, the simultaneous or delayed co-treatment with CB produced more pronounced effects on micronuclei frequency. Thus, based on our data, we may summarize that both tested variants of the protocol (without and with CB) can be used for genotoxicity testing. However, some differences in the results in relation to exposure time, doses and types of NMs with various properties could be observed. Additionally, a cell type used for testing can play a crucial role in results interpretation [60]. It is thus recommended that the standardised testing strategy is thoroughly verified in future studies, and the authors need to be aware of the fact that the different response to the NMs in various in vitro models can be influenced by various factors, including e.g., by the culture medium.

## Figures and Tables

**Figure 1 nanomaterials-11-01929-f001:**
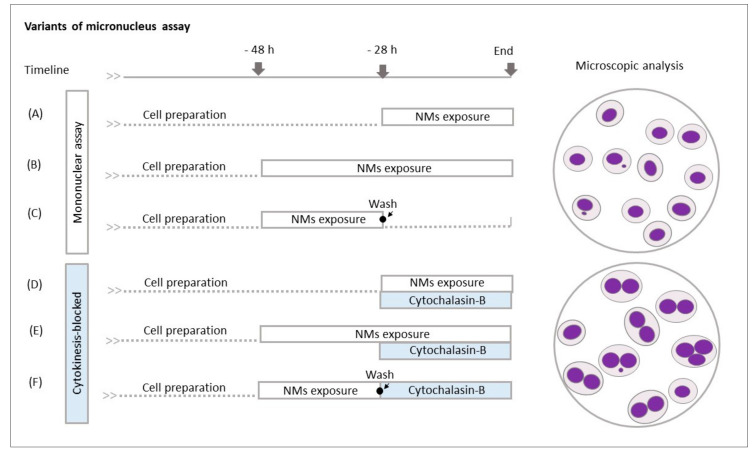
Overview of micronucleus assay variants considered in the study design. (**A**–**C**): Micronucleus assay without CB treatment where the mononucleated cells are scored: (**A**) A 28 h treatment with NM (“short time” treatment without CB); (**B**) a 48 h treatment with NM (“long time” treatment without CB); (**C**) a 20 h treatment with NM followed by 28 h post-cultivation after washing NM off from a sample (post-treatment cultivation without CB). (**D**–**F**): Cytokinesis-blocked micronucleus assay with CB treatment where the binucleated cells are scored: (**D**) A 28 h co-treatment with NM and CB (co-treatment with CB); (**E**) a 48 h treatment with NM [(20 h NM treatment followed by 28 h co-treatment with CB (delayed co-treatment with CB)]; (**F**) a 20 h treatment with NM followed by a 28 h treatment with CB after washing NM off (post-treatment CB). In this study, variants (**A**,**B**,**D**,**E**) were experimentally investigated, while variants (**C**,**F**) were not tested (see the text for details).

**Figure 2 nanomaterials-11-01929-f002:**
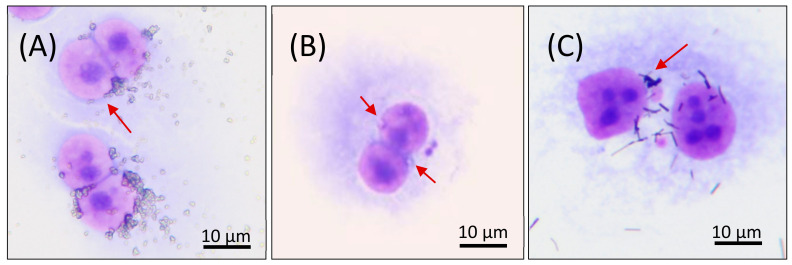
Microscopic analysis of BEAS-2B cells cultivated with various nanomaterials. Examples of binucleated cells without (**A**) or with MN (**B**,**C**) with tested NMs (red arrows) as detected by the optical microscope: (**A**) TiO_2_ (NM101), concentration 10 µg/mL, 28 h co-treatment with CB; (**B**) SiO_2_ (NM200), concentration 1 µg/mL, 48 h delayed co-treatment with CB; (**C**) Ag (NM302), concentration 25 µg/mL, 28 h co-treatment with CB.

**Figure 3 nanomaterials-11-01929-f003:**
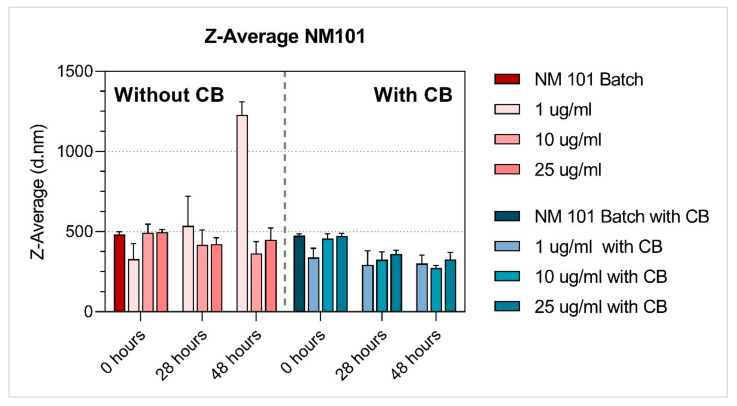
Average hydrodynamic size Z-Average [d.nm] of TiO_2_ NM101 with and without cytochalasin-B (CB). Dynamic light scattering (DLS) measurement of NM after preparation in diH_2_O containing 0.05% BSA (batch), and in BEGM™ culture medium 0 h, 28 h and 48 h after preparation. NM characterisation in BEGM™ culture medium was done with and without CB (1 µg/mL). The results are presented as the mean of repeated runs (12–24) per measurement.

**Figure 4 nanomaterials-11-01929-f004:**
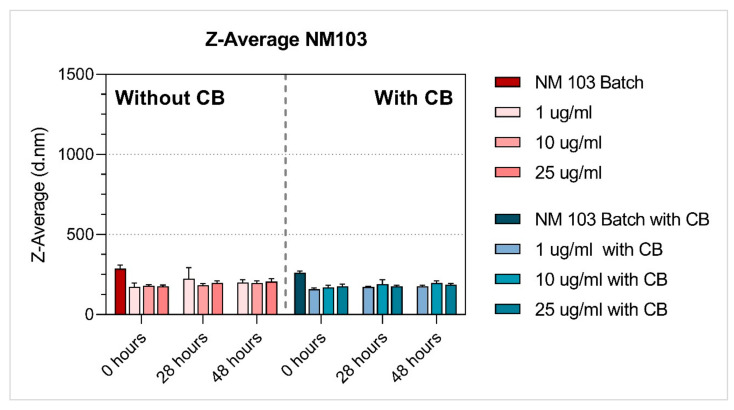
Average hydrodynamic size Z-Average [d.nm] of TiO_2_ NM103 with and without cytochalasin-B (CB). Dynamic light scattering (DLS) measurement of NM after preparation in diH_2_O containing 0.05% BSA (batch), and in BEGM™ culture medium 0 h, 28 h and 48 h after preparation. NM characterisation in BEGM™ culture medium was done with and without CB (1 µg/mL). The results are presented as the mean of repeated runs (12–24) per measurement.

**Figure 5 nanomaterials-11-01929-f005:**
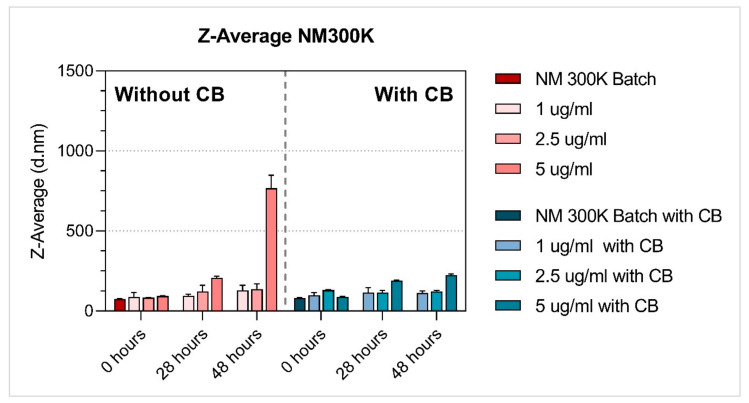
Average hydrodynamic size Z-Average [d.nm] of Ag NM300K with and without cytochalasin-B (CB). Dynamic light scattering (DLS) measurement of NM after preparation in diH_2_O containing 0.05% BSA (batch), and in BEGM™ culture medium 0 h, 28 h and 48 h after preparation. NM characterisation in BEGM™ culture medium was done with and without CB (1 µg/mL). The results are presented as the mean of repeated runs (12–24) per measurement.

**Figure 6 nanomaterials-11-01929-f006:**
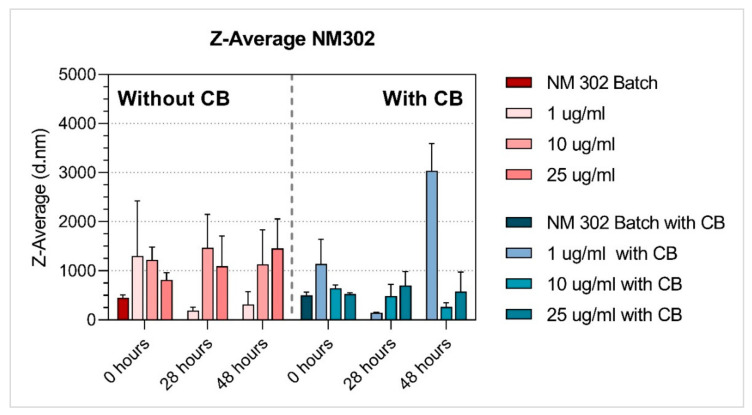
Average hydrodynamic size Z-Average [d.nm] of Ag NM302 with and without cytochalasin-B (CB). Dynamic light scattering (DLS) measurement of NM after preparation in diH_2_O containing 0.05% BSA (batch), and in BEGM™ culture medium 0 h, 28 h and 48 h after preparation. NM characterisation in BEGM™ culture medium was done with and without CB (1 µg/mL). The results are presented as the mean of repeated runs (12–24) per measurement.

**Figure 7 nanomaterials-11-01929-f007:**
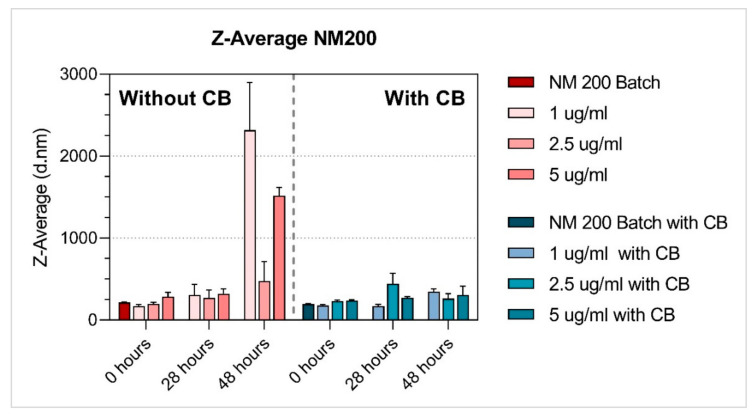
Average hydrodynamic size Z-Average [d.nm] of SiO_2_ NM200 with and without cytochalasin-B (CB). Dynamic light scattering (DLS) measurement of NM after preparation in diH_2_O containing 0.05% BSA (batch), and in BEGM™ culture medium 0 h, 28 h and 48 h after preparation. NM characterisation in BEGM™ culture medium was done with and without CB (1 µg/mL). The results are presented as the mean of repeated runs (12–24) per measurement.

**Figure 8 nanomaterials-11-01929-f008:**
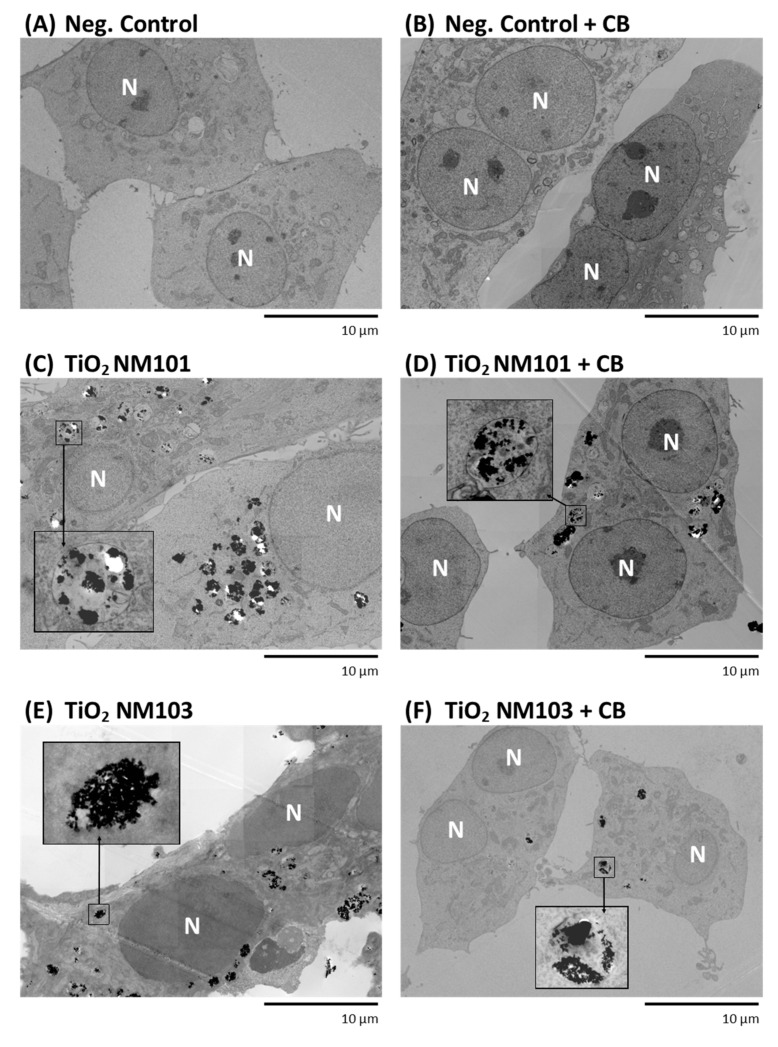
A Transmission electron microscopy images of nanomaterials in BEAS-2B cell line. The representative TEM images showing the tested NMs inside the BEAS-2B cells with various levels of aggregation and/or magnification (without cytochalasin-B (**A**,**C**,**E**), with cytochalasin-B (**B**,**D**,**F**)). Cell nucleus is marked “N”. (**A**) negative control; (**B**) negative control with cytochalasin-B; (**C**) titanium dioxide NM101; (**D**) titanium dioxide NM101 with cytochalasin-B; (**E**) titanium dioxide NM103; (**F**) titanium dioxide NM103 with cytochalasin-B. B Transmission electron microscopy images of nanomaterials in BEAS-2B cell line. The representative TEM images showing the tested NMs inside the BEAS-2B cells with various levels of aggregation and/or magnification (without cytochalasin-B (**G**,**I**,**K**), with cytochalasin-B (**H**,**J**,**L**)). Cell nucleus is marked “N”. (**G**) synthetic amorphous silicon dioxide NM200; (**H**) synthetic amorphous silicon dioxide NM200 with cytochalasin-B; (**I**) silver NM300K; (**J**) silver NM300K with cytochalasin-B; (**K**) silver rods NM302; (**L**) silver rods NM302 with cytochalasin-B.

**Figure 9 nanomaterials-11-01929-f009:**
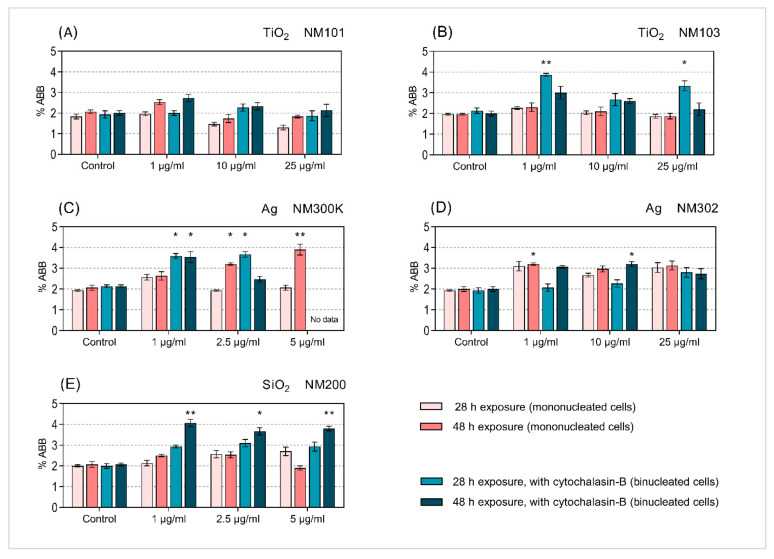
Frequencies of MN induced by exposure of BEAS-2B cells to selected NMs for 28 h and 48 h with and without CB. BEAS-2B cells that were exposed to selected NMs [TiO_2_ NM101 (**A**) and NM103 (**B**), SiO_2_ NM200 (**E**), Ag NM300K (**C**) and NM302 (**D**)] of specific size and shape for 28 h or 48 h. The results are presented as the mean percentage of aberrant cells (% ABB) in mononucleated or binucleated cells for assay without or with CB. Statistical analysis was done using two-proportion z-test to compare control groups versus treated groups (* *p* < 0.05; ** *p* < 0.01).

**Table 1 nanomaterials-11-01929-t001:** Selected NMs and their basic characteristics.

Nanomaterial		Details				References
Type of Material	NM Code	Shape	Size	Aggregation (Size Range)	Other Information	
TiO_2_	NM-101	Spherical or ellipsoidal	5 nm	10–170 nm 95.2% up to 100 nm 77.3% up to 50 nm	Anatase form	[23]
TiO_2_	NM-103	Ellipsoidal	20–100 nm	40–400 nm 51.8% up to 100 nm 12.7% up to 50 nm	Rutile form	[23]
SiO_2_	NM-200	Spherical or ellipsoidal	10–20 nm	15–650 nm 88.7% up to 100 nm 69.8% up to 50 nm	Precipitated	[24]
Ag	NM-300K	Round, triangular or trapezium	<20 nm	Low risk ^a^		[25]
Ag	NM-302	Rods	100–200 nm (width) 5–10 µm (length)	Aggregates of 5–10 particles (most common) up to thousands of particles (size > 10 µm) (rare)		[26]

^a^ Aggregates are observed in some cases connected with incorrect handling with NM vial.

**Table 2 nanomaterials-11-01929-t002:** Overview of the NM properties during their preparation in MilliQ-water containing 0.05% BSA (batch).

Nanomaterials			Properties (Mean ± SD)		
Type of Material	NM Code	Concentration	Hydrodynamic Size [nm]	Polydispersity Index	Zeta-Potential [mV]
TiO_2_	NM-101	2.56 mg/mL	481.82 ± 16.40	0.35 ± 0.04	n. m.
	NM-103	2.56 mg/mL	286.75 ± 21.69	0.36 ± 0.01	n. m.
SiO_2_	NM-200	2.56 mg/mL	214.07 ± 5.83	0.30 ± 0.03	n. m.
Ag	NM-300K	2.56 mg/mL	75.66 ± 0.81	0.28 ± 0.01	n. m.
	NM-302	2.56 mg/ml	451.07 ± 59.21	0.44 ± 0.11	n. m.

n. m.: not measured.

**Table 3 nanomaterials-11-01929-t003:** Overview of the NM properties during their cultivation in BEGM^TM^ culture medium without cytochalasin-B.

Nanomaterials			Properties								
Type of Material	NM Code	Conc.	Hydrodynamic Size [nm] (Mean ± SD)			Polydispersity Index (Mean ± SD)			Zeta-Potential [mV] (Mean ± SD)		
			0 h	28 h	48 h	0 h	28 h	48h	0 h	28 h	48 h
TiO_2_	NM-101	1 µg/mL	328.08 ± 97.08	535.05 ± 185.24	1226.57 ± 83.31	0.46 ± 0.06	0.56 ± 0.12	0.36 ± 0.08	−14.35 ± 0.66	−14.89 ± 1.36	−17.64 ± 0.81
		10 µg/mL	491.30 ± 54.83	417.43 ± 92.98	364.04 ± 72.42	0.37 ± 0.06	0.44 ± 0.06	0.46 ± 0.08	−13.43 ± 0.92	−12.35 ± 1.33	−13.10 ± 0.87
		25 µg/mL	496.59 ± 17.11	422.23 ± 38.99	449.18 ± 72.40	0.35 ± 0.02	0.49 ± 0.05	0.50 ± 0.06	−14.04 ± 1.48	−12.98 ± 1.14	−15.00 ± 1.63
TiO_2_	NM-103	1 µg/mL	173.38 ± 22.29	223.65 ± 69.07	201.06 ± 17.45	0.28 ± 0.08	0.26 ± 0.04	0.31 ± 0.06	−11.71 ± 0.97	−10.90 ± 1.16	−11.17 ± 0.79
		10 µg/mL	179.03 ± 7.69	184.20 ± 8.92	196.69 ± 12.91	0.25 ± 0.04	0.20 ± 0.06	0.25 ± 0.09	−11.65 ± 0.87	−12.39 ± 1.38	−12.70 ± 0.73
		25 µg/mL	176.80 ± 7.74	198.70 ± 10.20	203.22 ± 40.72	0.26 ± 0.04	0.26 ± 0.04	0.27 ± 0.05	−13.21 ± 1.11	−12.12 ± 0.73	−13.51 ± 1.38
SiO_2_	NM-200	1 µg/mL	169.12 ± 19.18	305.74 ± 131.54	2317.11 ± 581.67	0.47 ± 0.08	0.57 ± 0.09	**0.97 ± 0.06 ^a^**	−11.59 ± 1.27	−18.76 ± 2.31	−14.85 ± 1.56
		2.5 µg/mL	196.97 ± 18.18	269.07 ± 97.76	473.70 ± 238.19	0.44 ± 0.03	**0.61 ± 0.11 ^a^**	0.59 ± 0.08	−12.32 ± 1.32	−13.54 ± 1.65	−10.99 ± 2.75
		5 µg/mL	214.07 ± 5.83	321.40 ± 59.21	1518.18 ± 98.06	0.03 ± 0.03	0.55 ± 0.12	**0.65 ± 0.09 ^a^**	−12.98 ± 1.31	−12.95 ± 1.76	−12.87 ± 1.23
Ag	NM-300K	1 µg/mL	87.35 ± 28.13	95.78 ± 8.49	128.97 ± 31.27	0.37 ± 0.07	0.33 ± 0.07	0.46 ± 0.06	−12.43 ± 1.46	−12.29 ± 1.54	−13.71 ± 1.99
		2.5 µg/mL	83.38 ± 1.44	121.82 ± 38.25	137.31 ± 31.75	0.31 ± 0.03	0.31 ± 0.07	0.45 ± 0.08	−12.70 ± 1.34	−13.08 ± 1.59	−13.80 ± 2.16
		5 µg/mL	92.86 ± 2.51	208.28 ± 10.34	767.25 ± 81.01	0.28 ± 0.03	0.42 ± 0.02	**0.66 ± 0.12 ^a^**	−11.56 ± 1.07	−12.78 ± 1.53	−21.11 ± 1.64
Ag	NM-302	1 µg/mL	1305.16 ± 1118.92	191.39 ± 66.06	317.95 ± 259.75	**0.78 ± 0.25 ^a^**	0.53 ± 0.11	**0.60 ± 0.18 ^a^**	−14.20 ± 1.57	−13.07 ± 1.77	−13.31 ± 1.70
		10 µg/mL	1224.80 ± 261.61	1469.97 ± 679.78	1130.63 ± 702.44	**0.85 ± 0.14 ^a^**	**0.95 ± 0.09 ^a^**	**0.79 ± 0.24 ^a^**	−12.81 ± 0.86	−13.75 ± 1.15	−12.83 ± 1.13
		25 µg/mL	816.42 ± 148.07	1094.03 ± 611.9	1456.28 ± 599.52	**0.70 ± 0.09 ^a^**	**0.83 ± 0.16 ^a^**	**0.88 ± 0.12 ^a^**	−15.05 ± 1.50	−13.97 ± 1.04	−14.18 ± 1.80

^a^ According to the manufacturer, DLS is suitable only for suspension with polydispersity index < 0.7. Samples exceeding this limit recommendation are in bold.

**Table 4 nanomaterials-11-01929-t004:** Overview of the NM properties during their cultivation in BEGM^TM^ culture medium with cytochlasin-B.

Nanomaterials			Properties								
Type of Material	NM Code	Conc.	Hydrodynamic Size [nm] (mean ± SD)			Polydispersity Index (mean ± SD)			Zeta-Potential [mV] (mean ± SD)		
			0 h	28 h	48h	0 h	28 h	48h	0 h	28 h	48 h
TiO_2_	NM-101	1 µg/mL	338.52 ± 57.40	292.89 ± 87.35	301.13 ± 52.18	0.48 ± 0.06	0.43 ± 0.06	0.49 ± 0.08	−13.80 ± 1.32	−13.34 ± 1.11	−14.15 ± 1.78
		10 µg/mL	457.29 ± 29.64	324.67 ± 48.30	272.77 ± 15.09	0.35 ± 0.03	0.332 ± 0.07	0.29 ± 0.06	−12.03 ± 0.84	−12.02 ± 1.06	−13.86 ± 1.21
		25 µg/mL	472.23 ± 15.35	359.9 ± 2298	327.09 ± 42.38	0.29 ± 0.03	0.32 ± 0.05	0.36 ± 0.07	−13.18 ± 1.42	−13.49 ± 1.26	−13.55 ± 0.53
TiO_2_	NM-103	1 µg/mL	157.78 ± 9.13	171.43 ± 5.80	176.11 ± 6.40	0.25 ± 0.05	0.22 ± 0.02	0.21 ± 0.03	−12.13 ± 1.31	−11.74 ± 1.14	−12.62 ± 0.98
		10 µg/mL	170.34 ± 11.03	190.71 ± 27.24	198.16 ± 10.98	0.24 ± 0.02	0.22 ± 0.07	0.25 ± 0.05	−13.55 ± 0.91	−12.69 ± 1.06	−14.44 ± 1.09
		25 µg/mL	176.98 ± 12.53	177.29 ± 4.37	187.26 ± 7.08	0.27 ± 0.04	0.20 ± 0.04	0.22 ± 0.05	−12.83 ± 1.34	−13.29 ± 0.71	−13.41 ± 1.68
SiO_2_	NM-200	1 µg/mL	179.87 ± 9.75	171.54 ± 20.58	345.68 ± 33.06	0.48 ± 0.03	0.48 ± 0.13	**0.61 ± 0.15 ^a^**	−12.13 ± 1.66	−11.67 ± 0.80	−12.69 ± 1.84
		2.5 µg/mL	228.86 ± 15.24	443.91 ± 128.20	261.11 ± 60.00	0.45 ± 0.04	**0.61 ± 0.13 ^a^**	0.52 ± 0.07	−12.19 ± 1.62	−12.59 ± 1.07	−11.44 ± 1.01
		5 µg/mL	239.98 ± 7.19	271.98 ± 13.32	306.87 ± 103.28	0.39 ± 0.03	0.53 ± 0.08	0.49 ± 0.08	−11.33 ± 0.86	−12.42 ± 1.36	−11.64 ± 1.07
Ag	NM-300K	1 µg/mL	98.03 ± 16.15	115.68 ± 29.75	112.68 ± 11.32	0.31 ± 0.08	0.33 ± 0.05	0.33 ± 0.06	−12.34 ± 1.04	−9.54 ± 1.08	−10.39 ± 1.27
		2.5 µg/mL	130.07 ± 3.38	115.33 ± 12.71	122.95 ± 4.81	0.45 ± 0.04	0.29 ± 0.05	0.30 ± 0.04	−11.67 ± 1.48	−11.36 ± 1.23	−10.66 ± 1.53
		5 µg/mL	88.23 ± 3.68	188.85 ± 3.75	223.93 ± 8.09	0.30 ± 0.05	0.26 ± 0.01	0.38 ± 0.01	−13.05 ± 1.07	−11.65 ± 0.93	−9.94 ± 1.25
Ag	NM-302	1 µg/mL	1139.19 ± 500.85	142.20 ± 9.56	3038.32 ± 551.73	**0.84 ± 0.13 ^a^**	0.47 ± 0.08	**0.81 ± 0.09 ^a^**	−14.18 ± 0.99	−12.70 ± 0.90	−11.92 ± 0.88
		10 µg/mL	649.89 ± 63.78	485.20 ± 241.45	266.27 ± 81.81	0.63 ± 0.07	**0.68 ± 0.11 ^a^**	0.54 ± 0.13	−14.07 ± 0.84	−14.83 ± 1.37	−14.08 ± 0.80
		25 µg/mL	527.47 ± 26.76	704.09 ± 280.04	580.01 ± 398.89	0.47 ± 0.03	**0.68 ± 0.13 ^a^**	**0.59 ± 0.2 1^a^**	−14.03 ± 1.08	−15.01 ± 1.54	−14.71 ± 1.57

^a^ According to the manufacturer, DLS is suitable only for suspension with polydispersity index < 0.7. Samples exceeding this limit recommendation are in bold.

## Data Availability

The data presented in this study are available in the article and the Appendix A.

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
