# Peer review of "Testing Strategies of the In Vitro Micronucleus Assay for the Genotoxicity Assessment of Nanomaterials in BEAS-2B Cells"

_nanomaterials, 2021, doi:10.3390/nano11081929_

Round 1

Reviewer 1 Report

The paper is addressing a very important issue in genotoxicity testing of NMs, and contains a comprehensive dataset from testing of several groups of NMs with different variants of the MN assay protocol. It is essential information that these differences in protocol can be determining for if you get a positive or negative effect of the NMs tested. Validation of testing protocols for application with NMs is essential, and this paper contributes significantly to the field. Good that uptake analysis were performed by TEM. Nice to see all the detailed information included in section 2.2.

However, there are some major and minor comments to the paper that should  be addressed before it can be accepted for publication:

Major revisions:

  • In general, the discussion should be more focused and the results from other studies discussed in direct relation to the results from this study. Keep the discussion in line with the title of the study
  • The part with testing of LCH-9 medium is confusing and should be rewritten and better related to the results of the study.
  • The possible effect of CB on blocking of endocytosis should be discussed in relation to the results, as effect of CB was seen.
  • The aim of the study was to compare results from different experimental designs, and not primarily the effects of the different NMs tested. This is not reflected in the discussion part, which needs considerable improvements. The paper should in conclusion give recommendations for protocol for MN testing of NMs.
  • Information about positive controls missing
  • Figure 1 gives a good overview of the different micronucleus assay variants. However, Figure 1 should visualize only the different scenarios tested, and the possibility with washing step should be explained in the legend only, as well as why this was not included. Why use different time for NMs exposure in micronucleus variants D and F (28h and 20h)?
  • Figure 3: With CB, the size of NM101 seems to decrease with time after exposure. This should be commented/explained. How many different dispersions were measured for each concentration and time point? How can the big increase in size without CB at 48h be explained? No big increases in size are measured with CB – this should be explained.
  • Figure 4: how would you explain the reduction of size in media?
  • Effects of the different exposure times and with or without CB should be discussed better as well as recommendations for testing of NMs by MN.
  • are any differences between your protocol and the OECD guideline for in vitro micronucleus assay, with respect to analysis and calculations?
  • Size dependent effects should be discussed also in light of uptake
  • Effect of serum should be better discussed and referenced.
  • Possible interference between the NM and read-out of the assay should be addressed.
  • General comment: The tables present results by mean+-SD. For figures this information is lacking. Please also include information about number of replica and independent experiments in the table/figure caption.
  • Line 168: How many cells were seeded and exposed? Is the number high enough to get a reliable percentage of micronuclei in the population?

Minor revisions:

  • Line 57: This is not a definition but a recommendation for a definition, so the sentence could be rephrased starting for example with «Nanomaterials can be defined as (...) as suggested by (...) in 2011.»
  • Line 80-83: Long sentence, should be rephrased.
  • Line 96-98: These factors should be included in the discussion, which is important based on the title of the manuscript. Please explain why you chose 1 µg/ml CB.
  • Line 105: Unclear if the previously published data is produced by you or other authors.
  • Line 131: Missing information about concentration of NPs for the NMs available in dispersed form, and not stated if the calculation to dry NMs performed represent the amount of e.g Ag in the dispersion or total dispersion.
  • Line 134-136: Was the sonicator calibrated? How much energy is delivered?
  • Line 139: How long after preparation of stock dispersion were the NMs diluted in culture medium?
  • Line 142: DLS measurements of stock should be performed in same dispersion as applied in the experiments – with NMs in 0,05 % BSA solution, and not only in diH2O.
  • Line 179: solvent controls should be the same as dispersants – distilled water with 0.05% BSA. In addition, dispersant controls for the NMs available in dispersed form must be tested. This is not described clearly
  • Line 179-180: Please explain why you included DMSO and diH2O as controls.
  • Line 181: Unclear sentence. Did you perform three independent experiments with cells at the same passage, or one experiment with triplicate cultures (replicas)? If possible also include the number of passages used for the cells before experiment, in the section about cell culture.
  • Line 200: From the same experiment or from three independent experiments? (See previous comment).
  • Line 664: NM103 is TiO2 and not Ag as stated
  • Table 2: concentrations missing for TiO2 and Ag
  • Section 2.5 TEM: How many samples/experiments/number of cells were analyzed?
  • Section 3.3: The second paragraph has long sentences which makes it hard to read. Which control level do you refer to in line 279? Positive control?
  • Reference 27 and 28: information missing
  • Table 1: a unclear to me what is meant by information about aggregation for NM-302.
  • Table 2A: Please update the title to make it clear that the NMs are dispersed in water with BSA (0.05%) and not in both solutions separately. «Properties <1 h» should be rephrased to for example «Properties at preparation» or just «Properties» (as in B and C).
  • Table 2B and 2C: Please make the title more specific by including the name of the culture medium and the time of measurement (within how long after preparation of the solution).
  • Reference number 35: Please include URL or more information in the list.

Author Response

We thank the Reviewer for the valuable comments that helped to improve our manuscript. Please, find our response below and in the revised version of the manuscript.

Major revisions:

In general, the discussion should be more focused and the results from other studies discussed in direct relation to the results from this study. Keep the discussion in line with the title of the study

Response: The discussion was modified to better reflect the aims of our study.

The part with testing of LCH-9 medium is confusing and should be rewritten and better related to the results of the study.

Response: The text was modified as suggested.

The possible effect of CB on blocking of endocytosis should be discussed in relation to the results, as effect of CB was seen.

The aim of the study was to compare results from different experimental designs, and not primarily the effects of the different NMs tested. This is not reflected in the discussion part, which needs considerable improvements. The paper should in conclusion give recommendations for protocol for MN testing of NMs.

Response: These points are addressed in the modified discussion.

Information about positive controls missing

Response: BEAS-2B cells were also exposed to benzo[a]pyrene (B[a]P, 25-200 µM) and 1-nitropyrene (1-NP, 1-50 µM) for 28h and 48h. Positive results are not included, values are related to the negative control. The information on positive control is provided in section 2.4.

Figure 1 gives a good overview of the different micronucleus assay variants. However, Figure 1 should visualize only the different scenarios tested, and the possibility with washing step should be explained in the legend only, as well as why this was not included. Why use different time for NMs exposure in micronucleus variants D and F (28h and 20h)?

Response: As Reviewer 2 appreciated the figure, we would prefer to keep the original version. However, the figure legend was modified to make clear that not all MN variants were tested in our study.

Regarding differences in the exposure time, the testing variants are designed so that a total experimental time is 48 h (including all manipulations with the samples). Thus, for the variant C and F, the exposure was shortened to 20 h, although longer time could theoretically be used.

Figure 3: With CB, the size of NM101 seems to decrease with time after exposure. This should be commented/explained. How many different dispersions were measured for each concentration and time point? How can the big increase in size without CB at 48h be explained? No big increases in size are measured with CB – this should be explained.

Response: We speculated that CB may influence the NMs aggregation in cell culture medium. Thus, we proceeded with DLS measurement for both conditions. Smaller average hydrodynamic size (Z-Average) after application CB may be caused by different composition of NM protein corona, however, more detailed investigation is out of the scope of this study. We prepared 5 ml of each sample (e.g. NM101, 25 ug/l without CB) and at each timepoint we took 1 ml to measure DLS. One dispersion was measured for each concentration and timepoint.

Figure 4: how would you explain the reduction of size in media?

Response: NM103 (TiO2, rutile, primary size 20-100 nm, aggregation 40-400 nm) batch was measured in diH2O with 0.05% BSA at concentration of 2.56 mg/ml. Hydrodynamic size for batch under these conditions immediately after preparation was 286.75±21.69 nm. We explain the reduction of size in culture medium immediately after dilution as an effect of different dilution medium and lower concentration of NM (size range from 173.38±22.29 nm to 179.03±7.69 nm, dilution range 2560x - 102x). Various conditions may affect the size change, such as restricted diffusion, electrostatic repulsion, multiple scattering, and reversible self-association but this is not our field of expertise.

(https://www.malvernpanalytical.com/en/learn/knowledge-center/whitepapers/WP140207ApplicDLSprotein2)

Effects of the different exposure times and with or without CB should be discussed better as well as recommendations for testing of NMs by MN.

Response: These points are now covered in the revised discussion.

are any differences between your protocol and the OECD guideline for in vitro micronucleus assay, with respect to analysis and calculations?

Response: In this study, we used the same protocol (exposure time, number of evaluated BNC) as in our previous report (Cervena et al., Basic Clin. Pharmacol. Toxicol. 2016, 1–7.) with respect to the original OECD guidelines. In addition, a total of 3x 1000 mononucleated cells were also evaluated in protocol variants without CB (in agreement with the point 45 of the OECD guidelines).

Size dependent effects should be discussed also in light of uptake

Response: NPs uptake is mainly done via endocytosis (phagocytosis, clathrin-mediated endocytosis (CME), caveolae-dependent endocytosis, clathrin/caveolae independent endocytosis, micropinocytosis) or even passive diffusion, hole formation or electroporation. Cell type, as well as the proteins, lipids, and other molecules on cellular membrane or NPs influence the NPs uptake. For larger particles or agglomerates, phagocytosis is the most common way of NPs uptake. Smaller NPs clusters can be engulfed depending on their physico-chemical properties (eg. caveolae are 50–80 nm in size) and even diffusion can be involved. The key role is the physical interaction between NPs and cellular membrane.

Behzadi, S.; Serpooshan, V.; Tao, W.; Hamaly, M. A.; Alkawareek, M. Y.; Dreaden, E. C.; Brown, D.; Alkilany, A. M.; Farokhzad, O. C.; Mahmoudi, M. Cellular Uptake of Nanoparticles: Journey inside the Cell. Chem. Soc. Rev. 2017, 46 (14), 4218–4244. https://doi.org/10.1039/c6cs00636a.

Sushant Singh, Anh Ly, Soumen Das, Tamil S. Sakthivel, Swetha Barkam & Sudipta Seal (2018) Cerium oxide nanoparticles at the nano-bio interface: size-dependent cellular uptake, Artificial Cells, Nanomedicine, and Biotechnology, 46:sup3, S956-S963, DOI: 10.1080/21691401.2018.1521818

Effect of serum should be better discussed and referenced.

Response: BEAS-2B cells are grown in serum-free medium (BEGMTM), thus we did not compare results with and without medium although the presence of various proteins can change the protein corona of NPs. There are several articles mentioning the effect of serum on NMs, we added another reference to discussion.

Possible interference between the NM and read-out of the assay should be addressed.

Response: The influence of NM interference is mentioned in discussion. Higher amounts of NMs in cells can impair the visibility of MN.

General comment: The tables present results by mean+-SD. For figures this information is lacking. Please also include information about number of replica and independent experiments in the table/figure caption.

Response: The information was added to the description of Figure 3-7 and 9.

Line 168: How many cells were seeded and exposed? Is the number high enough to get a reliable percentage of micronuclei in the population?

Response: We seeded 6000 cells/well 24 hours before exposure (for 28 h exposure) and  3000 cells/well (for 48 h exposure) in order to avoid full confluency at time of examination. A total of 3x 500 binucleated cells and 3x 1000 mononucleated cells per each tested compound were evaluated.

Minor revisions:

Line 57: This is not a definition but a recommendation for a definition, so the sentence could be rephrased starting for example with «Nanomaterials can be defined as (...) as suggested by (...) in 2011.»

Response: The sentence was modified as suggested.

Line 80-83: Long sentence, should be rephrased.

Response: The sentence was split and modified.

Line 96-98: These factors should be included in the discussion, which is important based on the title of the manuscript. Please explain why you chose 1 µg/ml CB.

Response: Discussion was modified. 1 µg/ml of CB was chosen based on our previous work with BEAS-2B cells cultivated in BEGMTM medium (cytotoxicity and genotoxicity tests).

Line 105: Unclear if the previously published data is produced by you or other authors.

Response: The data published by us and other authors were meant. The sentence was modified.

Line 131: Missing information about concentration of NPs for the NMs available in dispersed form, and not stated if the calculation to dry NMs performed represent the amount of e.g Ag in the dispersion or total dispersion.

Response: Information was added to the text.

Line 134-136: Was the sonicator calibrated? How much energy is delivered?

Response: Ultrasonic homogenizer (S-450d, Branson, MO, USA) was calibrated according to NANOGENOTOX dispersion protocol (400 W, 10% amplitude, 3,136 MJ/m3). The information was added to the text.

Line 139: How long after preparation of stock dispersion were the NMs diluted in culture medium?

Response: For exposure and DLS measurement, NMs were diluted in culture medium immediately after the preparation and NMs were used within 1 hour. The information is now provided in the text.

Line 142: DLS measurements of stock should be performed in same dispersion as applied in the experiments – with NMs in 0,05 % BSA solution, and not only in diH2O.

Response: The DLS measurement of stock was performed in the same dispersion as in the experiments (diH2O containing 0.05% BSA). We never used only diH2O as it is not a part of NANOGENOTOX dispersion protocol. We added missing part about 0.05% BSA to the text.

Line 179: solvent controls should be the same as dispersants – distilled water with 0.05% BSA. In addition, dispersant controls for the NMs available in dispersed form must be tested. This is not described clearly

Response: The controls were used to identify basal MN frequency in the cells. BSA has no effects on MN induction, thus it was not added to diH2O. DMSO was used as a solvent for CB in binucleated samples.

Line 179-180: Please explain why you included DMSO and diH2O as controls.

Response: Please, see the previous response.

Line 181: Unclear sentence. Did you perform three independent experiments with cells at the same passage, or one experiment with triplicate cultures (replicas)? If possible also include the number of passages used for the cells before experiment, in the section about cell culture.

Response: The experiments were performed as triplicate cultures. The text was modified. Cell line BEAS-2B was obtained from ATCC® (CRL-9609TM, Manassas, VA, USA) at passage P38. For all tests we used P48.

Line 200: From the same experiment or from three independent experiments? (See previous comment).

Response: Please see the explanation above.

Line 664: NM103 is TiO2 and not Ag as stated

Response: Figure 2 caption was modified.

Table 2: concentrations missing for TiO2 and Ag

Response: The concentration was the same for all NM in the table. The table was modified.

Section 2.5 TEM: How many samples/experiments/number of cells were analyzed?

Response: TEM images were done in order to confirm/rule out the presence of NMs in cells. 3 samples per NM were analyzed in one experiment; number of cells was not specified. The text was modified.

Section 3.3: The second paragraph has long sentences which makes it hard to read. Which control level do you refer to in line 279? Positive control?

Response: The text in the paragraph was modified. We refer to the negative control. The text was modified.

Reference 27 and 28: information missing

Response: Information added to the list of references.

Table 1: a unclear to me what is meant by information about aggregation for NM-302.

Response: Small aggregates of 5-10 particles were most commonly formed for NM-302. Large aggregates exceeding 10 µm were rare. The text was modified.

Table 2A: Please update the title to make it clear that the NMs are dispersed in water with BSA (0.05%) and not in both solutions separately. «Properties <1 h» should be rephrased to for example «Properties at preparation» or just «Properties» (as in B and C).

Response: The title of the table was modified.

Table 2B and 2C: Please make the title more specific by including the name of the culture medium and the time of measurement (within how long after preparation of the solution).

Response: Culture medium specification was added to Table 2B and 2C. Samples labeled as 0h were measured in matter of minutes after preparation.

Reference number 35: Please include URL or more information in the list.

Response: URL was included

Reviewer 2 Report

General comments

In the manuscript “Testing strategies of the in vitro micronucleus assay for the genotoxicity assessment of nanomaterials in BEAS-2B cells” the authors performed a comparison of the genotoxic effects induced by five NMs with the application of four testing strategies of the MN assay, accompanied by confirmation of the presence of NMs inside the cells. They also  identify the conditions, which can possibly affect the interpretation of the genotoxicity results.

These topics are of great relevance due to the impact on environmental and human health, so this manuscript could provide important reference information that are of wide interest.

The manuscript is well written and the experimental design appears very clear. The authors have consulted a good number of scientific papers (57). I well appreciated the overview of micronucleus assay variants considered in the study design illustrated in Figure 1. Also, the figure 8 is of high quality.

Specific comments

Some points of discussion are to be rewritten, for increasing in value the paper, also in relation to recent literature.

Author Response

We thank the Reviewer for the positive comments. We modified the manuscript as requested.

The manuscript is well written and the experimental design appears very clear. The authors have consulted a good number of scientific papers (57). I well appreciated the overview of micronucleus assay variants considered in the study design illustrated in Figure 1. Also, the figure 8 is of high quality.

Specific comments

Some points of discussion are to be rewritten, for increasing in value the paper, also in relation to recent literature.

Response: Discussion was modified as requested.